# Potential Utility of Ultrasound-Enhanced Delivery of Antibiotics, Anti-Inflammatory Agents, and Nutraceuticals: A Mini Review

**DOI:** 10.3390/antibiotics11101290

**Published:** 2022-09-22

**Authors:** J. Karim Ead, Arjun Sharma, Miranda Goransson, David G. Armstrong

**Affiliations:** 1Keck School of Medicine, University of Southern California (USC), Los Angeles, CA 90089, USA; 2California University of Science and Medicine (CUSM), Colton, CA 92324, USA; 3HCA Florida Westside Hospital, Plantation, FL 33324, USA

**Keywords:** antibiotics, phonophoresis, podiatry, ultrasound, wound healing, wound management, transdermal drug delivery, low-intensity ultrasound phonophoresis, pulsed and continuous ultrasound

## Abstract

Ultrasound technology has therapeutic properties that can be harnessed to enhance topical drug delivery in a process known as phonophoresis. The literature on this method of drug delivery is currently sparse and scattered. In this review, we explore in vivo and in vitro controlled trials as well as studies detailing the mechanism of action in phonophoresis to gain a clearer picture of the treatment modality and explore its utility in chronic wound management. Upon review, we believe that phonophoresis has the potential to aid in chronic wound management, particularly against complicated bacterial biofilms. This would offer a minimally invasive wound management option for patients in the community.

## 1. Introduction

Ultrasound (US) technology utilizes the piezoelectric effect to generate inaudible high-frequency pressure waves [1]. These waves can be transferred through a coupling medium (e.g., gel) onto the surface of tissue [2]. The diagnostic imaging utility of US technology is well known. However, US can also be utilized in a non-invasive manner to help facilitate subcutaneous drug delivery. Since the pioneering work of Fellinger and Schmid, physical therapists have been using phonophoresis to manage chronic pain and enhance anti-inflammatory drug delivery [3]. Fellinger and Schmid demonstrated successful treatment of polyarthritis of the hand by driving hydrocortisone ointment into the inflamed area with ultrasound [3]. Additionally, phonophoresis has also been studied in the treatment of suppurative wounds [4]. Levenets et al. found that the phonophoresis of ethylenediaminetetraacetic acid with the quinoxaline antibiotic dioxidine (2,3-di- quinoxaline~-1,4-dioxide) was effective in accelerating wound healing by eliminating necrotic tissue [1]. It should be noted that there is currently limited evidence in the literature supporting the practice of phonophoresis, and the efficacy for therapeutic use is still in question. Many of the studies examined in this systematic review include a number of methodological shortcomings. Controlled clinical studies are rare and many questions remain unanswered, including the actual depth of the medication penetration, appropriate concentration of the medication, type of medium, ultrasound frequency, and ultrasound mode (continuous or pulsed). Therefore, the purpose of this literature review is to assess recent literature and to expand on the mechanism of action and potential benefit of phonophoresis.

## 2. Phonophoresis Mechanism of Action

Phonophoresis (PH) is the process of increasing skin absorption and penetration of topical medications to deep tissues using US [2]. PH replaces the traditional coupling agents (i.e., gel) used in US with a drug (i.e., topical antibiotics, anti-inflammatories) to be delivered [1,2]. The efficacy of this drug delivery modality is influenced by many parameters of US as well as the characteristics of the tissue it acts upon [2]. The literature places emphasis on two parameters of US: the power (expressed in Watts) and the frequency (expressed in MHz) [5]. While the exact mechanism of how PH enhances tissue permeability has not been fully elucidated, there have been many theories in the literature ranging from thermal to mechanical effects. Here, we will focus on the most well accepted primary mechanism of PH: a process known as cavitation [4,5]. Cavitation is the result of a natural process known as rectified diffusion [6]. In acoustic fields, such as those created by US, oscillating pressure waves cause gas bubbles that already exist in the field to undergo expansion and contraction [6]. This growth and collapse of bubbles or pockets on the skin’s surface and within the lipid bilayers of the stratum corneum temporarily disrupt the structure of the skin and allow for enhanced skin permeability [6]. Thus, creating an environment favoring the penetration of the topical agent [6]. 

In 2011, Polat et al. added an important clarification to the literature on how high frequency PH (HFP) and low frequency PH (LFP) can influence the process of cavitation [3]. With HFP defined as roughly 1 MHz to 3 MHz, the acoustically activated bubbles tend to be on a smaller scale (~1 um) and therefore cavitation can occur within the skin layers, particularly the lipid bilayers of the stratum corneum [3]. However, with LFP on the order of 20 kHz to 60 kHz, the acoustically activated bubbles are much larger (~150 um) and so cavitation occurs above the skin within the coupling medium of choice [3]. Therefore, choice of coupling medium is important for LFP [3]. Besides the mechanical effects ultrasound has on enhancing drug delivery, it has been proposed that this modality promotes anti- inflammatory effects which may include increased fibroblast recruitment, accelerated angiogenesis, increased matrix synthesis, and increased collagen fibril density [3]. These factors harmonize the complex nature of wound healing [3]. Due to its anti-inflammatory and membrane permeability enhancing properties, US could be taken beyond simple transdermal drug delivery and can also be used to help combat wounds infested with biofilms [7]. 

## 3. Wound Wars: Return of the Biofilm!

In recent years, some have speculated that biofilm could be an essential culprit in the pathogenicity of chronic wounds [8]. Biofilms have been established and found to form on various medical devices (urinary catheters, orthopedic implants). Biofilm is a dynamic community of microbes with an array of genetic diversity and gene expression that projects unique defense and behavior mechanisms that could serve as the fuel source of chronic infections [9]. Electron microscopy of biopsies from chronic wounds discovered that 60% of the specimens contained biofilm structures [10], whereas biofilm in acute wounds only comprised 6% of the total microbial population [10].

How do biofilms form? (See Figure 1).

Free floating planktonic bacteria initially disperse but then eventually (reversibly) attach to surface structures [11].However, if the planktonic species start to multiply, they become more firmly attached (sessile) and differentiate, consequently changing their genetic makeup to promote survival. This complex communication mechanism is known as quorum sensing [12].Once the aggregated planktonic species reach an optimal level, they form an extracellular polymeric substance (EPS) [13].The EPS generally is composed of various polysaccharides, proteins, glycolipids and bacterial DNA that essentially functions as a protective barrier [13].

Additionally, once biofilms maturate, they become progressively resistant to antimicrobial therapies [13,14,15,16,17]. It is important to note that debridement is essential to not only stimulate cellular communication but to also disrupt biofilm formation. However, the physical elimination of biofilm, clinical and in vitro models have established that debridement opens a time-dependent window during which applied topical treatments can suppress biofilm reformation [13]. Many studies have also shown the efficacy of changing cellular microenvironments in the permeation of biofilms [14,15,16,17].

This scenario can be better illustrated by following Figure 2. Once a patient has clinical features that raise suspicion of a biofilm, sharp surgical debridement is often the next step in management. As mentioned above, this helps to disrupt the biofilm complicating the wound and provides a time dependent window to reassess the use of antibiotics to treat the wound. Ultrasound can fit into this treatment plan as a less invasive alternative to sharp debridement that would concurrently deliver antibiotics to prevent biofilm reformation. A recent literature review conducted by Kataoka et al. found that US debridement using non-contact devices improved wound healing by attenuating inflammatory responses [18].

## 4. Materials and Methods

The keywords “antibiotic phonophoresis”, “therapeutic ultrasound”, “ultrasound guided drug delivery”, and “phonophoresis in wound management” were searched through the PubMed and Google Scholar search engines. Both in vivo and in vitro experimental trials were included in our search. We did not robustly assess study quality for inclusion criteria as the literature available on this topic was already sparse to begin with. Therefore, we included all trials we were able to identify.

## 5. Results

A summary of the collected studies can be found in the table below (Table 1). 

### 5.1. Phonophoresis with Topical Antibiotics

Although the literature is sparse, antibiotic PH has shown promising results between in vivo and in vitro studies [8,9,10,11,12,13,18]. According to the National Health Institute, 80% of human bacterial infections are associated with biofilms; these nefarious bacterial communities promote antimicrobial resistance to systemic and topical antibiotic therapies [19,20]. In order to effectively disrupt biofilm in chronic wounds, mechanical/physical debridement in conjunction with topical antimicrobial therapy is necessary. Thus, making phonophoresis an ideal treatment modality due to its dual action properties. In a case study by Ansari et al., erythromycin PH was performed on a 31-year-old woman with a 7-month history of chronic rhinosinusitis (CRS) [21]. While the patient′s previous drug therapies had been ineffective, erythromycin PH to both maxillary sinuses led to complete resolution of the patients’ symptoms, with normal sinuses seen CT scan and complete resolution of her symptoms on 5 month follow up [21]. In a larger human trial by Chen et al., isoniazid and rifampin phonophoresis were explored in the treatment of tuberculosis lymphadenitis patients [22]. Among forty-one patients, those that received rifampin phonophoresis saw a significant increase in skin absorption compared to transdermal drug delivery alone [22]. Isoniazid did not see similar success in absorption, but the authors attributed this to differences in biochemical properties between the two drugs, with Rifampin being more lipophilic [18]. This suggests that the efficacy may be drug dependent. 

Looking at in vitro studies, Horsely et al. studied the phonophoresis of gentamicin on a human urothelial organoid model for the treatment of urinary tract infections (UTIs) [19]. Specifically, this study evaluated the use of US-activated microbubbles (MB) filled with gentamicin against the uropathogen E. faecalis, which is a common culprit in chronically infected patients [19]. The phonophoresis gentamicin treatment resulted in a significant reduction in the MIC value compared to gentamicin alone [19]. With a fraction of the clinically approved dosage, US and gentamicin together showed similar bactericidal activity and a 75% reduction in bacterial burden when compared to free gentamicin treatment [19]. In another in vitro study, Dong et al. used vancomycin PH against Staph epidermidis biofilms to better understand how PH is effective against biofilms [23]. Upon treatment with US and vancomycin loaded MBs, the bacteria were seen to increase their permeability to extracellular material. Supporting this data, the post-treatment biofilms were more sensitive to vancomycin and showed a significantly reduced biomass compared to vancomycin treatment alone [23]. 

### 5.2. Phonophoresis with Topical Anti-Inflammatory Agents

Phonophoresis has also shown promising results when coupled with anti-inflammatory agents [20,21,22,23,24]. In a human trial by Cagnie et al., the efficacy of ketoprofen PH was explored in patients with knee disorders requiring arthroscopy [24]. Among 29 patients enrolled in the study, those that were given continuous or pulsed US treatment were shown to have significantly higher levels of ketoprofen in synovial tissue than topical ketoprofen application alone [24]. Saliba et al. found similar results when testing the phonophoresis of dexamethasone onto the anterior forearm of 10 healthy patients [25]. Their results revealed a higher serum concentration of dexamethasone with US compared to a negligible serum concentration seen with dexamethasone alone [25]. These enhanced drug delivery effects do appear to have a tangible symptomatic benefit for patients experiencing pain, as demonstrated by Luksurpan et al. [26]. In their 2013 study, 23 out of 46 patients were given piroxicam PH to treat knee osteoarthritis [26]. Based on the Western Ontario and McMaster Universities Osteoarthritis Index (WOMAC) and the Kellgren-Lawrence Grade I to III, all patients treated with PH saw a significantly greater reduction in pain and improvement in knee function [25]. Supporting these results, a study by Ay et al. showed that PH of diclofenac was seen to significantly improve pain and neck function in patients suffering from myofascial pain syndrome [27]. 

Animal studies using anti-inflammatory agent PH have also shown good results. In a study carried out by Cardoso et al., the effect of diclofenac PH on paw edema and inflammatory mediators in rat models was evaluated in rat models [28]. 

The rats treated with PH saw a significantly greater reduction in paw edema as well as a significant reduction in inflammatory infiltration on histological analysis [28]. In an older animal study by Davick et al., 15 dogs were given various concentrations of cortisol cream either with or without US [29]. Those that received 10% cortisol cream with US had significantly higher penetration than those that used cortisol cream alone [29]. It is worth noting that human trials using PH of anti-inflammatory agents have yielded mixed results. Klaiman et al. studied phonophoresis compared to ultrasound therapy alone in 49 patients with various soft tissue injuries in a randomized, double-blinded, uncontrolled trial [30]. Each group underwent treatments 3 times a week for 3 weeks [30]. Both treatment groups had decreased pain levels at the end of 3 weeks, but there were no significant difference between the treatment groups [30]. Similarly, Kozanoglu et al. studied the effects of ibuprofen phonophoresis versus traditional solitary ultrasound therapy in patients with knee osteoarthritis [31]. They found that both modalities were generally well tolerated and effective but that there was no significant difference in improvement rates between the two groups [31].

### 5.3. Phonophoresis with Using Nutraceuticals 

In addition to more traditional anti-inflammatory agents, nutraceuticals have also been evaluated for their potential clinical utility when delivered via ultrasound. A recent study analyzed wound healing via garlic extract PH in twenty-four male albino rats [32]. Upon histologic analysis, both garlic and PH helped wound healing, but garlic extract PH revealed faster and better (i.e., more complete wound) healing compared to garlic extract application without PH [32]. This can potentially be explained due to garlic’s anti-inflammatory and ultrasound induced biofilm disruption (garlic/ultrasound) [32]. Filho et al. compared topical use of Aloe vera gel, pulsed mode US and Aloe vera phonophoresis on rat paw with collagenase-induced tendinitis [33]. Edema size, tensile tendon strength, tendon elasticity, number of inflammatory cells and tissue histology were studied at 7 and 14 days after tendinitis induction [33]. Topical application of Aloe vera gel did not show any statistically significant improvement in the inflammatory process, whereas phonophoresis enhanced the gel action reducing edema and number of inflammatory cells, promoting the rearrangement of collagen fibers and promoting also the recovery of the tensile strength and elasticity of the inflamed tendon to recover their normal pre-injury status [33]. 

**Table 1 antibiotics-11-01290-t001:** Current Literature on Phonophoresis.

Study	Sample Size	Sample Characteristics	Frequency	US Type and Intensity	Topical Agent	Duration of Treatment
Ansari et al. (2013) [21]	1	Animal	1 MHz	Pulsed, 1.0 W/cm^2^	Erythromycin	5 min, every other day for 10 total treatment sessions
Ay et al. (2010) [27]	20	Animal	1.0 MHz	Not Specified, 1.5 W/cm^2^	Diclofenac	10 min, 5 times a week over 3 weeks, 15 total treatment sessions
Cagnie et al. (2003) [24]	20	Animal	1.0 MHz	Pulsed & Continuous, 1.5 W/cm^2^	Ketoprofen	5 min
Fares et al. (2017) [32]	6	Animal	1.0 MHz	Pulsed, 1.5 W/cm^2^	Garlic Extract	5 min, 3 treatment sessions a week for 3 weeks
Saliba et al. (2007) [25]	10	Human	3.0 MHz	Pulsed, 1.0 W/cm^2^	Dexamethasone	5 min
Horsely et al. (2019) [19]	3	Human	1.1 MHz	Pulsed with MBs, 2.5 Mpa	Gentamicin	20 s
Dong et al. (2017) [23]	3	Human	1.0 MHz	Pulsed with MBs, 0.5 W/cm^2^	Vancomycin	5 min
Cardoso et al. (2019) [28]	66	Human	1.0 MHz	Pulsed, 1.0 W/cm^2^	Diclofenac	1 min
Chen et al. (2016) [22]	41	Human	1.0 MHz	Not Specified, 0.75 mW/cm^2^	Isoniazid and Rifampin	30 min
Filho et al. (2010) [33]	20	Human	1.0 MHz	Pulsed, 0.5 W/cm^2^	Aloe Vera	2 min, for 7 days
Davick et al. (1988) [29]	9	In Vitro	0.87 MHz	Not Specified, 0.5 mW/cm^2^	Cortisol Cream	8 min
Luksurapan et al. (2013) [26]	23	In Vitro	1.0 MHz	Continuous, 1.0 W/cm^2^	Piroxicam	10 min, 5 times a week, for 2 weeks

## 6. Discussion

Due to its promising therapeutic effects, phonophoresis has been employed in various clinical settings such physical therapy, sports medicine, and wound management among others. Once again, a full summary of the works we explored in this review can be found in Table 1. There are few randomized, controlled clinical trials documenting the effectiveness of phonophoresis as it pertains to the management of chronic wounds. However, it stands to reason that this modality may have a synergy with antibiotics against biofilms by disrupting their EPS and allowing antimicrobials to effectively target bacterial species. It should be noted that phonophoresis is not without limitations. For one, many of the studies cited in Table 1 would have patients return for treatments multiple times a week. Such a time-consuming process could pose a patient compliance issue if not streamlined. In addition, there may be some potential for skin irritation around wounds due to the thermal effects of ultrasound.

With the current development of US based therapeutic approaches, the hope is that this technology may be transferred into clinical practice to treat a wide variety of conditions. It should be noted that clinicians should always evaluate local and potential systemic effects from phonophoresis. More research is needed to clarify the ultrasound parameters that are critical for maximizing local diffusion of topically applied drugs with sonation and how to measure and control the local vs. systemic effects of phonophoresis. We look forward to future works that can confirm or refute this approach. 

## 7. Conclusions

Phonophoresis does have the potential to aid the current treatment modalities against bacterial biofilms in chronic wounds. While more research is needed to confirm the appropriate indications for its use and efficacy, the current body of literature shows promise.

## Figures and Tables

**Figure 1 antibiotics-11-01290-f001:**
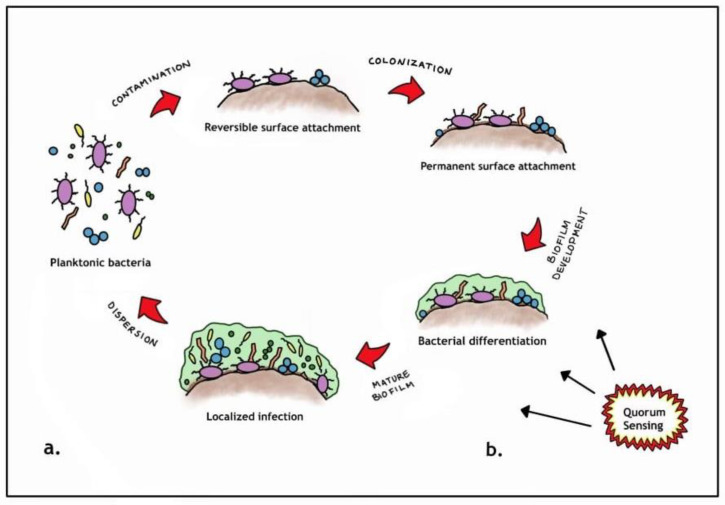
Miranda Goransson’s Pictorial biofilm pathogenesis.

**Figure 2 antibiotics-11-01290-f002:**
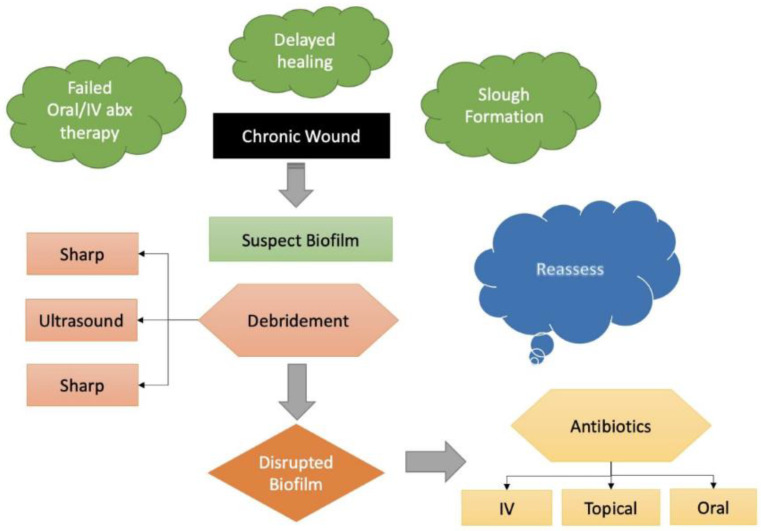
Biofilm Treatment Algorithm. Once a biofilm is suspected, debridement should be performed to disrupt the biofilm and improve the utility of post-debridement antibiotics. A less invasive alternative to sharp debridement could be US.

## Data Availability

Not applicable.

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
