# Peer review of "Potential Utility of Ultrasound-Enhanced Delivery of Antibiotics, Anti-Inflammatory Agents, and Nutraceuticals: A Mini Review"

_antibiotics, 2022, doi:10.3390/antibiotics11101290_

Round 1

Reviewer 1 Report

The review article entitled: Potential Utility of Ultrasound-Enhanced Delivery of Antibiotics, Anti-Inflammatory Agents, and Nutraceuticals by the authors Karim Ead et,al., have done a great job by determining the  Potential Utilization of Ultrasound-Enhanced Delivery of Antibiotics, Anti-Inflammatory Agents, and Nutraceuticals. There are some flaws that shall be addressed .some of them are:
1.The title of the manuscript  must be changed as:

Potential Utility of Ultrasound-enhanced delivery of antibiotics, anti-Inflammatory agents and Nutraceutical: A Mini Review.

2.In abstract line 10 the word  harseneed may please be replaced by  word strapped.

3.In abstract add a few lines regarding the role of Ultrasound-Enhanced Delivery of Antibiotics in community.

4. Add some lines for the searching sources  in the methodology section in sketch form too.

5. Add conclusion of the review in a few lines at the end for the authors easy understandability..

6. Yet I am not a native speaker of English language but still I recommend that the English language needs touching up in a major way. The article needs to be rewritten in readable English. Many sentences are confusing, do not lead to scientific meaning, and can be found starting in 

Author Response

  1. The title of the manuscript must be changed as:

Potential Utility of Ultrasound-enhanced delivery of antibiotics, anti-Inflammatory agents and Nutraceutical: A Mini Review. 

Thank you for this suggestion. The manuscript title has now been changed to “Potential Utility of Ultrasound-enhanced delivery of antibiotics, anti-Inflammatory agents and Nutraceutical: A Mini Review.” 

  1. In abstract line 10 the word  harseneed may please be replaced by word strapped.

Thank you for this suggestion. However, we feel that the word harnessed would better capture the meaning of our sentence in this line. We mean to use harness in the sense of making use of the natural properties of ultrasound rather than straps. We apologize for any confusion. 

  1. In abstract add a few lines regarding the role of Ultrasound-Enhanced Delivery of Antibiotics in community.

We agree with this and have incorporated your suggestion into the manuscript. Line 16 of our abstract now reads: “This would offer a minimally invasive wound management option for patients in the community.” 

  1. Add some lines for the searching sources in the methodology section in sketch form too.

The search terms can be found in section 4 of the manuscript. 

  1.  Add conclusion of the review in a few lines at the end for the authors easy understandability.

Thank you for this suggestion. We have incorporated the following conclusion into the manuscript “Phonophoresis does have the potential to aid the current treatment modalities against bacterial biofilms in chronic wounds. While more research is needed to confirm the appropriate indications for its use and efficacy, the current body of literature shows promise.” 

  1. Yet I am not a native speaker of English language but still I recommend that the English language needs touching up in a major way. The article needs to be rewritten in readable English. Many sentences are confusing, do not lead to scientific meaning, and can be found starting in.

We appreciate you giving us this feedback. We will review the whole manuscript and try our best to make sure it is as clear as possible, however, it is difficult for us to know what changes to make without specific examples. 

Reviewer 2 Report

In this paper, the authors summarized the utility of ultrasound in the delivery of antibiotics, anti-inflammatory agents, and nutraceuticals. But the explanation of the mechanism of phonophoresis in drug delivery is not comprehensive and precise enough. In my opinion, the manuscript is not recommended for publication in its current form. My main concerns are as follows: 1) Mechanisms of action of ultrasound are ranging from thermal to mechanical effects. However, in this paper, the authors only focus on cavitation and lack an explanation of thermal and mechanical effects; 2) The description and explanation of Figure 2 are poor. In order to improve the readability of the paper, the biofilm treatment algorithm should be explained and demonstrated in the manuscript; 3) In the discussion section, it is recommended to summarize the existing works and emphasize the novelty and limitations of ultrasound technology in clinical application; 4) Modify the reference styles in line 64 and line 158; 5) The following references are recommended for how to address biofilm permeation; Small 2021, 2101495, ACS Appl. Mater. Interfaces 2020, 12, 22479; Chemical Engineering Journal 426 (2021) 131919; Nano Today, 2022, 46, 101602.

Author Response

  1. Mechanisms of action of ultrasound are ranging from thermal to mechanical effects. However, in this paper, the authors only focus on cavitation and lack an explanation of thermal and mechanical effects

Thank you for pointing this out. While it is true that there are multiple proposed mechanisms of action for phonophoresis, cavitation is the most well accepted primary mechanism for phonophoresis, which is why we chose to focus on it. We have added a clarification for this focus so that the manuscript now reads “While the exact mechanism of how PH enhances tissue permeability has not been fully elucidated, there have been many theories in the literature ranging from thermal to mechanical effects. Here, we will focus on the most well accepted primary mechanism of PH: a process known as cavitation [4,5]. Cavitation is the result of a natural process known as rectified diffusion [6].” 

  1. The description and explanation of Figure 2 are poor. In order to improve the readability of the paper, the biofilm treatment algorithm should be explained and demonstrated in the manuscript

We agree with this and have incorporated your suggestion into the manuscript. This additional explanation for Figure 2 has been added into Section 3: “This scenario can be better illustrated by following Figure 2. Once a patient has clinical features that raise suspicion of a biofilm, sharp surgical debridement is often the next step in management. As mentioned above, this helps to disrupt the biofilm complicating the wound and provides a time dependent window to reassess the use of antibiotics to treat the wound. Ultrasound can fit into this treatment plan as a less invasive alternative to sharp debridement that would concurrently deliver antibiotics to prevent biofilm reformation.” In addition, we have added to the description of Figure 2 so that it now reads,Figure 2. Biofilm Treatment Algorithm. Once a biofilm is suspected, debridement should be performed to disrupt the biofilm and improve the utility of post-debridement antibiotics. A less invasive alternative to sharp debridement could be US.” 

  1.  In the discussion section, it is recommended to summarize the existing works and emphasize the novelty and limitations of ultrasound technology in clinical application

Thank you for this suggestion. We have summarized the existing in Table 1 and have now emphasized this in the discussion. In addition, we have incorporated the following on the limitations of this treatment: “It should be noted that phonophoresis is not without limitations. For one, many of the studies cited in Table 1 would have patients return for treatments multiple times a week. Such a time-consuming process could pose a patient compliance issue if not streamlined. In addition, there may be some potential for skin irritation around wounds due to the thermal effects of ultrasound.” 

  1. Modify the reference styles in line 64 and line 158

Thank you for pointing this out. We have corrected the reference styles in these lines to be consistent with the other references. 

  1. The following references are recommended for how to address biofilm permeation; Small2021, 2101495, ACS Appl. Mater. Interfaces2020, 12, 22479; Chemical Engineering Journal 426 (2021) 131919; Nano Today, 2022, 46, 101602. 

Thank you for gathering these sources for us. We have incorporated these references into the manuscript as references 14-17. We also added the following address of the references in our biofilm section: “Many studies have also shown the efficacy of changing cellular microenvironments in the permeation of biofilms [14-17].” 

Reviewer 3 Report

First of all, I want to appreciate this great work. But I want to focus on some items regarding the following:

1) Title:

The title is completely different from the main aim of the study

The title is ‘potential Utility of Ultrasound-Enhanced Delivery of Antibiotics, Anti-Inflammatory Agents, and Nutraceuticals:” while the main aim of the study is “ we explore in vivo and in vitro controlled trials as well as studies detailing the mechanism of action in phonophoresis to gain a clearer picture of the treatment modality and explore its utility in chronic wound management.”

2)    The main aim of the paper is to give an overview of the utility of ultrasound in wound management through the literature, the authors mentioned the use of anti-inflammatories in OA ………… etc., not directed to wound management.

3)  In the introduction part;

The authors discussed the effect of ultrasound to eliminate necrotic tissue concentrating on the mechanical effect of ultrasound and without mentioning the phonophoretic effect.

In short, the authors went far from the main concept of this paper due to focusing on the effect of ultrasound and dropping the phonophoretic effect.

4)    In reference (17), authors refer to the treatment of chronic wounds, then mentioned a study applied to rhinosinusitis patients ….. and so, in references 18 and 19.

5)    Using the same references many times in the literature, however, the open search can supply the authors with many research papers regarding the main idea of the targeted subject.

6)    Repletion of the same references within the same idea 26,27, ……….. etc. while the available literature is numerous.

7)    Authors should explain the sites of application of phonophoresis i.e., the peripheries of the wound or the wound bed.

8)    Some additional keywords can be as

a)    Transdermal drug delivery

b)    Low-intensity ultrasound phonophoresis

c)    Pulsed and continuous ultrasound.

9)    Line (205), there are few randomized, controlled clinical trials ……….. the literature is rich in these kinds of research papers.

10)                    It is not acceptable to apply ultrasound on the wound bed for the purpose of phonophoresis as phonophoresis means transdermal drug delivery, while the skin is lost in cases of the wound, and so, why do we accept the idea of transferring drugs through the lost skin especially the crucial role of stratum corneum in absorption and penetration of drugs which is lost in all of wounds. 

Author Response

  1. Title:

The title is completely different from the main aim of the study 

The title is ‘potential Utility of Ultrasound-Enhanced Delivery of Antibiotics, Anti-Inflammatory Agents, and Nutraceuticals:” while the main aim of the study is “ we explore in vivo and in vitro controlled trials as well as studies detailing the mechanism of action in phonophoresis to gain a clearer picture of the treatment modality and explore its utility in chronic wound management.” 

Thank you for pointing this out. The manuscript title has now been changed to “Potential Utility of Ultrasound-enhanced delivery of antibiotics, anti-Inflammatory agents and Nutraceutical: A Mini Review.” 

  1. The main aim of the paper is to give an overview of the utility of ultrasound in wound management through the literature, the authors mentioned the use of anti-inflammatories in OA ………… etc., not directed to wound management.

Thank you for this suggestion. While we do agree that some of the papers we discussed do not pertain to wound management specifically, they are more so discussed to better establish the role ultrasound has played in enhancing drug delivery.  

  1.   In the introduction part;

The authors discussed the effect of ultrasound to eliminate necrotic tissue concentrating on the mechanical effect of ultrasound and without mentioning the phonophoretic effect. 

In short, the authors went far from the main concept of this paper due to focusing on the effect of ultrasound and dropping the phonophoretic effect. 

Thank you for pointing this out. While we do agree that the main purpose of this paper is to establish the phonophoretic effects of ultrasound, we do find it relevant to mention all properties of ultrasound in wound management. Even in the case of phonophoresis, all properties of ultrasound, including its mechanical effects, will be present and therefore can influence outcomes. 

  1. In reference (17), authors refer to the treatment of chronic wounds, then mentioned a study applied to rhinosinusitis patients ….. and so, in references 18 and 19.

Thank you for pointing this out. It is true that references 17-19 (now references 21-23 post-edit) are not directly related to the management of chronic wounds with ultrasound. Due to the limited number of in vivo studies specifically addressing chronic wound management and antibiotic phonophoresis, we had to look to phonophoresis used in other settings to support our claim. Chronic rhinosinusitis is a condition complicated by bacterial biofilms. As such, the efficacy of phonophoresis in this setting may be translatable to setting of chronic wounds, which are also complicated by bacterial biofilms. 

  1. Using the same references many times in the literature, however, the open search can supply the authors with many research papers regarding the main idea of the targeted subject.

See comment 9. 

  1. Repletion of the same references within the same idea 26,27, ……….. etc. while the available literature is numerous.

See comment 9. 

  1. Authors should explain the sites of application of phonophoresis i.e., the peripheries of the wound or the wound bed.
  2.   Some additional keywords can be as a)    Transdermal drug delivery b)    Low-intensity ultrasound phonophoresis c)    Pulsed and continuous ultrasound.

Thank you for this suggestion. We have added these keywords to our manuscript.  

  1. Line (205), there are few randomized, controlled clinical trials ……….. the literature is rich in these kinds of research papers.

Thank you for your feedback. While it is true that therapeutic ultrasound has numerous papers written on it, we find not specifically find controlled clinical trials for its use chronic wounds with antibiotics. However, we do agree that we vague about this in the manuscript and have now clarified this line to read: “There are few randomized, controlled clinical trials documenting the effectiveness of phonophoresis as it pertains to the management of chronic wounds.”  

  1. It is not acceptable to apply ultrasound on the wound bed for the purpose of phonophoresis as phonophoresis means transdermal drug delivery, while the skin is lost in cases of the wound, and so, why do we accept the idea of transferring drugs through the lost skin especially the crucial role of stratum corneum in absorption and penetration of drugs which is lost in all of wounds.

Thank you for bringing up this point. While we do agree that transdermal drug delivery is not applicable to wounds where the dermal layer has been compromised, we are focused on the problem of trans-biofilm drug delivery. If ultrasonic effects can disrupt a layer of cells like the stratum corneum, why can a similar effect not be applied to biofilm membranes surrounding bacteria?  

Round 2

Reviewer 1 Report

Acept after checking the manuscript for grammar and spellings.

Reviewer 2 Report

The authors have well addressed my question, and the paper could be considered for publication now.
